# Changes in Central Sensitivity to Thyroid Hormones vs. Urine Iodine during Pregnancy

**DOI:** 10.3390/medsci12040050

**Published:** 2024-09-27

**Authors:** Ioannis Ilias, Charalampos Milionis, Maria Alexiou, Ekaterini Michou, Chrysi Karavasili, Evangelia Venaki, Kostas Markou, Irini Mamali, Eftychia Koukkou

**Affiliations:** 1Department of Endocrinology, Diabetes and Metabolism, Elena Venizelou Hospital, GR-11521 Athens, Greece; pesscharis@hotmail.com (C.M.); mariaalexioumd@gmail.com (M.A.); katerina.michoy@yahoo.com (E.M.); chrysakaravasili@yahoo.gr (C.K.); evivenaki@gmail.com (E.V.); ekoukkou@gmail.com (E.K.); 2Division of Endocrinology, Department of Internal Medicine, School of Health Sciences, University of Patras, GR-26504 Patras, Greece; markoukonst@upatras.gr (K.M.); irenemamali@gmail.com (I.M.)

**Keywords:** central sensitivity, thyroid hormones, pregnancy, hypothalamic–pituitary–thyroid (HPT) axis, free thyroxine (fT4), iodine intake, thyroid feedback quantile-based index (TFQI), urinary iodine excretion (UI)

## Abstract

Introduction/Aim: Central sensitivity to thyroid hormones refers to the responsiveness of the hypothalamic–pituitary–thyroid (HPT) axis to changes in circulating free thyroxine (fT4). Although dose–response relationships between thyroid hormones per se and urinary iodine (UI) levels have been observed, central sensitivity to thyroid hormones in relation to UI remains unexplored. The aim of the present study was to evaluate central sensitivity to thyroid hormones (by means of the Thyroid Feedback Quantile-based Index [TFQI], which is a calculated measure, based on TSH and fT4, that estimates central sensitivity to thyroid hormones) in pregnancy and to assess whether it differs according to gestational age and/or iodine intake. Materials and Methods: One thousand, one hundred and two blood and urine samples were collected from pregnant women (with a mean age ± SD of 30.4 ± 4.6 years) during singleton pregnancies; women with known/diagnosed thyroid disease were excluded. Specifically, TSH and fT4, anti-thyroid peroxidase antibodies and UI were measured in each trimester and at two months postpartum, while the TFQI was calculated for all the study samples. After the elimination of outliers, statistical analysis was conducted with analysis of variance (ANOVA) for the variables versus time period, while Pearson’s correlation was used to assess the TFQI versus UI. Results: The mean TFQI index ranged from −0.060 (second trimester) to −0.053 (two months postpartum), while the corresponding UI was 137 and 165 μg/L, respectively. The TFQI-UI correlation was marginally negative (Pearson r: −0.323, *p*: 0.04) and significantly positive (r: +0.368, *p*: 0.050) for UI values over 250 μg/L, in the first and the second trimesters of pregnancy, respectively. Discussion: The TFQI is a new index reflecting central sensitivity to thyroid hormones. A lower TFQI indicates higher sensitivity to thyroid hormones. In our sample, the TFQI was mainly positively related to iodine intake in the second trimester of pregnancy (following the critical period of organogenesis). Thus, the observed changes in the TFQI may reflect the different ways of the central action of thyroid hormones, according to the phase of pregnancy. These results have the potential to enhance our comprehension of the changes in the HPT axis’ function via variations in central sensitivity to thyroid hormones and its interplay with nutritional iodine status during pregnancy.

## 1. Introduction

Iodine is the most important component of the thyroid hormone’s structure and function, playing a vital role in the synthesis of thyroxine (T4) and triiodothyronine (T3), which are key regulators of metabolism. Each molecule of T4 contains four iodine atoms, while T3, the more metabolically active form, contains three. The synthesis of these hormones begins with the active transport of iodide from the bloodstream into the thyroid gland’s follicular cells via the sodium-iodide symporter (NIS). Once inside the thyroid, iodide is oxidized and incorporated into the amino acid tyrosine on thyroglobulin to form monoiodotyrosine (MIT) and diiodotyrosine (DIT) [1]. These iodinated tyrosines couple to form T4 and T3, which are then released into circulation and taken up by peripheral tissues. Within these tissues, deiodinase enzymes modulate thyroid hormone activity by converting T4 into T3 or into the inactive reverse T3 (rT3). This regulation process is necessary for maintaining thyroid hormone homeostasis and adapting to metabolic needs. In pathological conditions such as hypothyroidism, hyperthyroidism, and non-thyroidal illness syndrome (NTIS), deiodinase expression and activity can be significantly altered. For instance, NTIS often involves increased type 3 deiodinase activity, enhancing T4 conversion to rT3 and reducing the availability of active T3. Consequently, the dysregulation of iodine metabolism and deiodinase function alters the balance required for thyroid hormone synthesis and action [1]. Pregnancy induces several physiological adaptations in the maternal thyroid axis to ensure adequate thyroid hormone levels for both the mother and the developing fetus. One of the key changes is an increase in the production of thyroxine-binding globulin (TBG) due to elevated estrogen levels, leading to a rise in total circulating thyroid hormone levels, though the free hormone levels typically remain within normal ranges [2]. Additionally, the demand for iodine may increase during pregnancy because of the requirements of the growing fetal thyroid (additionally, increased renal iodine clearance has been postulated in pregnancy, although this has been contested [3,4]. To meet iodine demand, the maternal thyroid enhances its ability to capture and concentrate iodine through the sodium-iodide symporter (NIS). Human chorionic gonadotropin (hCG), produced abundantly by the placenta, also stimulates the thyroid gland due to its weak thyroid-stimulating hormone (TSH)-like activity, often causing a transient decrease in maternal TSH levels during the first trimester. These adaptations help to maintain adequate free T3 and T4 levels, which are essential for fetal neurodevelopment and metabolic regulation. Iodine deficiency during pregnancy can impair fetal development, underscoring the importance of thyroid hormone regulation during this critical period [2]. Thus, during pregnancy, the thyroid undergoes dynamic changes to meet the increased demand for thyroid hormones, crucial for maternal and fetal health [5,6,7,8]. Adequate iodine intake is essential for thyroid hormone synthesis, particularly during this period of heightened metabolic demand [5,6,7,8]. Both TSH and free thyroxine (fT4) are subject to regulation through a negative feedback mechanism within the hypothalamic–pituitary–thyroid (HPT) axis. The concept of central sensitivity (the sensitivity of the HPT axis) to thyroid hormones, incorporating both fT4 and TSH levels, has been proposed as a functional measure of this axis [9]. Recent research works conducted in China have honed on the Thyroid Feedback Quantile-based Index (TFQI; a calculated parameter that is used to assess central sensitivity to thyroid hormones) in pregnancy, aiming to understand gestational complications. In the context of pre-pregnancy obesity, and gestational diabetes mellitus (GDM), the TFQI showed contrasting associations between obese and non-obese women [10]. The TFQI has stood out as an independent risk factor for fetal macrosomia in euthyroid pregnant women [11]. Elevated TFQI levels were associated with increased neonatal thyroid-stimulating hormone (TSH), suggesting a potential risk factor for congenital hypothyroidism [12]. Additionally, elevated TFQI levels during the first half of pregnancy has been linked to a lowered risk of GDM, possibly indicating a protective effect against this complication [8]. Nevertheless, the course of the TFQI during the second and third trimester of pregnancy, to the best of our knowledge, has not been evaluated.

Thyroid function tests are generally not considered to be sensitive enough to gauge iodine status in most populations. The relationship between fT4 levels and iodine intake is complex: although dose–response relationships between thyroid hormones per se and urinary iodine (UI) levels have been observed [12,13], the sensitivity to thyroid hormones in relation to UI remains unexplored. This study aimed to evaluate central sensitivity to thyroid hormones in pregnancy and assess whether it differs according to gestational age and/or to iodine intake.

## 2. Materials and Methods

This work is part of a larger ongoing protocol focused on assessing iodine nutritional status in pregnancy in Greece, following our earlier work on the subject [14]. The present study was retrospective, in which 1102 blood and urine spot samples were collected from pregnant women (with a mean age ± SD of 30.4 ± 4.6 years) in the three trimesters of pregnancy and at two-months postpartum (we considered the trimesters by gestational week as follows: 0 to 13th gestational week—first trimester; 14th to 26th week—second trimester; and the 27th week to the gestational term—third trimester). The inclusion criterion included singleton pregnancy in women aged 18 and older, while the exclusion criterion was medical history of thyroid disease and/or use of thyroid hormone before, during, or after pregnancy. This study utilized de-identified data to ensure anonymity. Standard clinical practices were followed without any additional interventions. The study subjects’ specimens (convenience sampling) were collected from all over Greece, as in a previous study [14], from public health care facilities (outpatient gynecology/endocrine clinics). Forty percent of samples were from the Athens capital area, while the others were from urban areas in Macedonia (northern Greece), Epirus (northeastern Greece), Thessaly (central Greece), Peloponnese (southern continental Greece), the Aegean islands (eastern Greece), the Ionian islands (western Greece), and Crete (southern Greece). All the samples were obtained from women who were Caucasian, with uncomplicated singleton pregnancies, with no known history of thyroid disease or receipt of any treatment for such disease or iodine supplements. For TSH, FT4 and anti-thyroid peroxidase antibodies (anti-TPO; considered as being positive at levels over 34 U/mL), morning blood samples were obtained at a single session along with a morning spot urine sample; all were stored at −20 °C until analysis. Blood tests were performed with electrochemiluminescence (Elecsys assays/Cobas E-411; Roche, Basel, Switzerland). For the determination of urine iodine (UI), ammonium persulfate was used to digest the samples, and the UI concentration was determined indirectly with a spectrophotometric method based on the Sandell–Kolthoff reaction, as outlined in our previous studies [14,15]. At least two measurements were performed for each sample, from which the mean value was calculated. All UI determinations were performed in the Endocrinology Laboratory of the Endocrinology Department, the Faculty of Medicine, the University of Patras, Greece. This laboratory participates in the EQUIP (Ensuring the Quality of Iodine Procedures, CDC, Atlanta, GA, USA; https://www.cdc.gov/laboratory-quality-assurance/php/inorganic-elements/equip.html?CDC_AAref_Val=https://www.cdc.gov/labstandards/equip.html) (accessed on 24 September 2024) international quality control program for iodine determinations. The TFQI was calculated to assess central sensitivity to thyroid hormones as follows: TFQI = cdfFT4 − (1 − cdfTSH); here, cdf is the cumulative distribution function of the parameter [8]. Positive TFQI values indicate lower sensitivity to thyroid hormones, while negative values indicate higher sensitivity.

Tukey’s inter-quartile range (IQR) “fence” method was used to indicate and exclude possible outliers in the UI values [16] (Figure 1). For the statistical analysis of TSH, fT4, UI and the TFQI by each time period, analysis of variance (ANOVA, with Tukey’s/Kramer’s post hoc testing) was used; the rate of anti-TPO positivity by trimester was assessed with the Chi square test; for UI by anti-TPO positivity and trimester, two-way ANOVA was used; and Pearson’s correlation was used to assess TSH, FT4, and the TFQI versus UI. Analysis was conducted mainly with JASP (Version 0.18.3; [Computer software]; JASP Team 2024, University of Amsterdam, Nieuwe Achtergracht 129B, Amsterdam, The Netherlands). For sensitivity analysis of our results, we also conducted a bootstrapped correlation analysis of the TFQI vs. UI after 1000 iterations [using the Resampling Procedures software (v.1) of Prof. Howell, D.C., University of Vermont, 2007; available at https://www.uvm.edu/~statdhtx/StatPages/Resampling/Resampling.html, accessed on 24 September 2024]. The Elena Venizelou Hospital’s Ethics Committee/Scientific Governance Council reviewed and approved the study protocol (No. 18/2019). All participants provided informed consent prior to their inclusion in this study.

## 3. Results

After the implementation of Tukey’s “fence” method for the elimination of values that could be considered as outliers, analysis was conducted on 1075 samples (27 samples were eliminated). Thyrotropin varied by time, being lower in the first trimester (*p*: 0.005) (Table 1), whereas FT4, the TFQI, and UI showed no significant change (Table 1). There were significant changes in anti-TPO antibodies by trimester (Table 1).

In 63% of measurements, UI fell below the minimum recommended value of 150 µg/L, while 34% of measurements were below 100 µg/L. Only 29% exhibited an optimal iodine level ranging from 150 µg/L to 250 µg/L. A significant correlation was noted between FT4 and UI in the first trimester of pregnancy (r: +0.095, *p*: 0.05); other correlations for UI with TSH and fT4 were not significant. Additionally, there was no association between anti-TPO positivity and UI.

The TFQI vs. UI correlation was negative and marginal (Pearson r: −0.323, *p*: 0.093) for UI values over the threshold of 250 μg/L in the first trimester. In contrast, a positive and significant correlation was found in the second trimester for UI values exceeding 250 μg/L (r: +0.368, *p*: 0.050) (Figure 2).

The bootstrapped correlation analysis of the TFQI vs. UI (for UI values above 250 μg/L), performed after 1000 iterations, yielded an r-value of −0.130 (95% CI: −0.459 to +0.183) for the first trimester and an r-value of +0.387 (95% CI: +0.024 to +0.667) for the second trimester.

## 4. Discussion

In this work, we studied the TFQI in pregnancy, along with UI. While the TFQI did not exhibit significant changes by trimester, the TFQI-UI correlation was marginally negative and significantly positive for UI values over 250 μg/L in the first and the second trimester of pregnancy, respectively. We also found expected changes in TSH and anti-TPO positivity during pregnancy, with most women having UI levels below the recommended range [17].

The TFQI is a new index reflecting central sensitivity to thyroid hormones. The latter has been associated with various health conditions, including obesity, metabolic syn-drome, impaired renal function, diabetes, and diabetes-related mortality [18,19,20,21,22,23,24,25]. A lower TFQI indicates higher sensitivity to thyroid hormones, and in our sample, in the second trimester (after the critical period of organogenesis), it was positively related to iodine intake, albeit over the normal threshold of UI values. These observed changes in the TFQI may reflect the different ways of central action of thyroid hormones according to the phase of pregnancy.

Pregnancy induces physiological alterations in thyroid hormone metabolism to meet increased demands. In our study, the relationship of the maternal TFQI with UI was significantly positive for UI levels higher than 250 μg/L only in the second trimester of pregnancy. At present, among the possible explanations, we can hypothesize that the observed correlation is the result of differences in the central expression/activity of deiodinases (particularly type 2) [26] across the time phases of pregnancy. Experimental research has shown that type 2 deiodinase expression in the pituitary of rats and dams is inhibited with high iodine intake [27,28]. Another possible explanation for the observed correlation could involve the gradual function of the fetal thyroid and its integration into the maternal HPT axis. The fetal thyroid begins to accumulate iodine by 10–12 weeks of gestation [29] and starts to produce hormones between 16 and 20 weeks of gestation [30]. Pregnant women need to increase their iodine intake to meet both maternal and fetal needs, and inadequate iodine intake prompts the maternal thyroid to produce more T3 and less T4 to conserve iodine [31]. The inverse may apply with abundant iodine intake and, coupled with fetal thyroid function, may be at the root of the noted TFQI vs. UI association. With progressing pregnancy (and changes in needs for thyroid hormones), the physiological adaptation of the HPT may reset the TFQI vs. UI association to be neutral or negative for UI exceeding 250 μg/L. Another possible factor, which influences the maternal and fetal HPT axes in pregnancy, is the placenta. Placental iodine clearance and placental thyroid hormone metabolism/deiodinase activity [14,32,33] may be also implicated in this adaptation of the HPT vis-à-vis UI.

The nutritional iodine status in Greece is suboptimal, partly due to the absence of a state-run iodization program. Since the 1960s, table sea salt has been iodized voluntarily by producers. No iodine supplements are routinely given to pregnant women. Despite earlier claims that Greece was iodine-sufficient [34], recent findings [14], including those from the present study, indicate that a significant proportion of pregnant women in Greece are iodine-deficient.

To acknowledge the limitations of our study, we have to note the lack of data on the Body Mass Index (BMI), weight change in pregnancy, TSH, fT4, and UI before pregnancy. Data on pregnancy complications were also not available. A more nuanced statistical analysis would be preferable for our data; however, there are caveats: not all subjects had assessments for more than one trimester, and accessible data on exact gestational age in weeks were currently available for a minority of women. Thus, we treated each measurement as being a separate one. The relationship between the TFQI and UI can be confounded by several characteristics of the participants (such as maternal age and smoking); a more complex analysis with linear mixed models, for example, could even adjust for such covariates.

## 5. Conclusions

Optimal maternal thyroid function contributes to a favorable intrauterine environment for fetal thyroid development and function. Adequate iodine intake, reflected by urine iodine levels, is crucial for supporting maternal and fetal thyroid hormone synthesis during pregnancy. The observed association of central sensitivity with the thyroid hormone, as evaluated by the TFQI, and iodine nutritional status, as determined by UI, over UI levels of 250 μg/L highlights the significance of optimal iodine intake vis-à-vis HPT status in pregnancy. The calculation of the TFQI, as reported in the manuscript, could serve as an ancillary parameter for assessing iodine nutritional status during pregnancy and identifying pregnant women who may require iodine supplementation. The calculation of the TFQI necessitates preferably large-sized data, so that the position within the distribution of all the values in this dataset for a subject’s TSH and FT4 can be pinpointed. Thus, a tool, like a visual guide or a nomogram, using the TFQI could be created to assess the probability of iodine adequacy in pregnancy. Additionally, assessing the TFQI with iodine intake in pregnant women could help to determine the appropriate iodine dosage for each trimester in conjunction with monitoring FT4 and TSH levels. Further relevant research is warranted to elucidate the underlying mechanisms driving this association and explore potential benefits or hazards and practical uses.

## Figures and Tables

**Figure 1 medsci-12-00050-f001:**
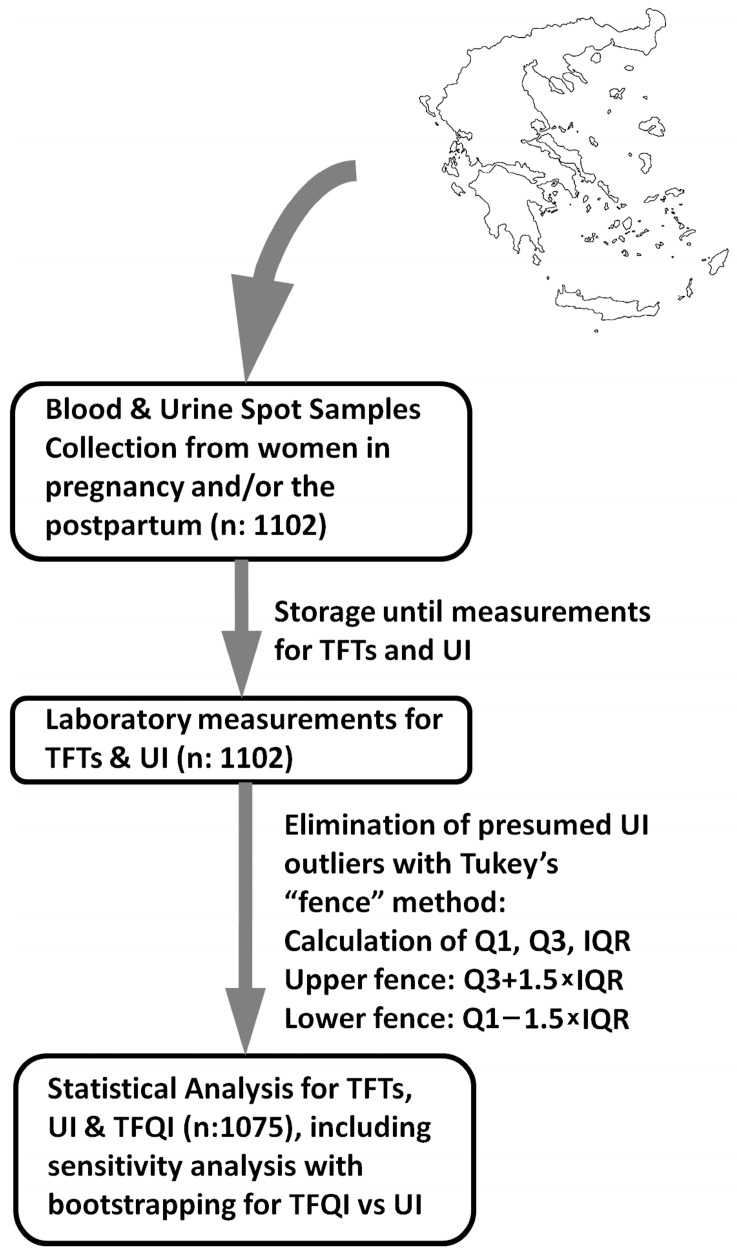
A flow chart of the study; TFT: thyroid function tests (and anti-thyroid peroxidase antibodies); UI: urine iodine; TFQI: Thyroid Feedback Quantile-based Index; Q1: first quartile; Q3: third quartile; IQR: inter-quartile range.

**Figure 2 medsci-12-00050-f002:**
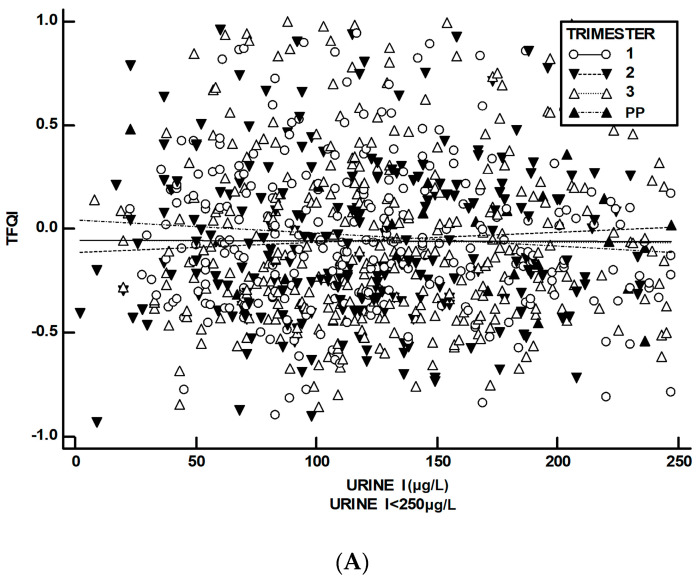
Scatter plots with ordinary least squares (OLS) regression lines for the TFQI vs. urine iodine (UI) levels up to 250 μg/L (**A**) and above 250 μg/L (**B**) during each trimester and the postpartum period (PP). Note: In the postpartum period, only one UI measurement exceeded 250 μg/L, precluding OLS regression for this time period.

**Table 1 medsci-12-00050-t001:** Thyroid function parameters and urinary iodine.

Time	First Trimester	Second Trimester	Third Trimester	Postpartum
N	427	268	355	25
TSH (μIU/mL)	1.56 ± 1.21 *	1.77 ± 0.98	1.77 ± 0.87 *	1.31 ± 0.98
FT4 (ng/dL)	1.31 ± 0.49	1.22 ± 0.59	1.22 ± 0.63	1.26 ± 0.33
FT4 (pmol/L)	16.86 ± 6.31	15.70 ± 7.59	15.70 ± 8.10	16.22 ± 4.25
Percentage positive anti-TPO (U/mL)	8%	6%	3% **	12%
TFQI	−0.056 ± 0.338	−0.060 ± 0.398	−0.059 ± 0.408	−0.053 ± 0.252
Urinary iodine (μg/L)	133 ± 68	137 ± 71	137 ± 65	165 ± 69

* ANOVA—Tukey/Kramer post hoc test *p*: 0.005, ** Chi Square *p*: 0.030.

## Data Availability

Data are available from Zenodo at https://doi.org/10.5281/zenodo.11488920.

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
