# Peer review of "Changes in Central Sensitivity to Thyroid Hormones vs. Urine Iodine during Pregnancy"

_medsci, 2024, doi:10.3390/medsci12040050_

Round 1

Reviewer 1 Report

Comments and Suggestions for Authors

In the introduction section, it could be important to describe that iodine is part of the thyroid hormone structure. Also, explain how tissues capture iodine and the link between thyroid hormones and their deiodinases in a pathological condition. Another point to be mentioned is the adaptation that pregnancy induces in the mother to maintain thyroid hormone levels.

In the method section, it is necessary to mention other independent variables such as age, number of pregnancies, prior thyroid illness, body weight, other pregnancy complications, etc. What were the inclusion and exclusion criteria?

In the figure 2, what is PP? Considering low and high levels of iodine, the correlation between iodine concentration and TSH or FT4 could be done. Which variable has the best correlation with iodine? TSH, FT4, or TFQI

Consider discussing the limitations of the study.

Author Response

Responses to Reviewers’ comments

We thank the reviewers for their comments and suggestions. Please see below for our detailed report regarding the revised version of our manuscript

[Reviewer 1].
Comments and Suggestions for Authors
[1]. In the introduction section, it could be important to describe that iodine is part of the thyroid hormone structure.

Thank you for the suggestion – Please see our response to comment [2] below.

[2]. Also, explain how tissues capture iodine and the link between thyroid hormones and their deiodinases in a pathological condition.

Thank you for the suggestion. We have added the following in the Introduction:

“Iodine is the most important component of thyroid hormone structure and function, playing a vital role in the synthesis of thyroxine (T4) and triiodothyronine (T3), which are key regulators of metabolism. Each molecule of T4 contains four iodine atoms, while T3, the more metabolically active form, contains three. The synthesis of these hormones begins with the active transport of iodide from the bloodstream into the thyroid gland’s follicular cells via the sodium-iodide symporter (NIS). Once inside the thyroid, iodide is oxidized and incorporated into the amino acid tyrosine on thyroglobulin to form monoiodotyrosine (MIT) and diiodotyrosine (DIT) [1]. These iodinated tyrosines couple to form T4 and T3, which are then released into circulation and taken up by peripheral tissues. Within these tissues, deiodinase enzymes modulate thyroid hormone activity by converting T4 into T3 or into the inactive reverse T3 (rT3). This regulation process is necessary for maintaining thyroid hormone homeostasis and adapting to metabolic needs. In pathological conditions such as hypothyroidism, hyperthyroidism, and non-thyroidal illness syndrome (NTIS), deiodinase expression and activity can be significantly altered. For instance, NTIS often involves increased type 3 deiodinase activity, enhancing T4 conversion to rT3 and reducing the availability of active T3. Consequently, dysregulation of iodine metabolism and of deiodinase func-tion alters the balance required for thyroid hormone synthesis and action”

[3]. Another point to be mentioned is the adaptation that pregnancy induces in the mother to maintain thyroid hormone levels.
Thank you for the suggestion. We have added the following in the Introduction, after the addition detailed in point [2]:
“Pregnancy induces several physiological adaptations in the maternal thyroid axis to ensure adequate thyroid hormone levels for both the mother and the developing fetus. One of the key changes is an increase in the production of thyroxine-binding globulin (TBG) due to elevated estrogen levels, leading to a rise in total circulating thyroid hor-mone levels, though the free hormone levels typically remain within normal ranges. Additionally, the demand for iodine may increase during pregnancy because of the requirements of the growing fetal thyroid (additionally increased renal iodine clear-ance has been postulated in pregnancy, although this has been contested. To meet iodine demand, the maternal thyroid enhances its ability to capture and concentrate iodine through the sodium-iodide symporter (NIS). Human chorionic gonadotropin (hCG), produced abundantly by the placenta, also stimulates the thyroid gland due to its weak thyroid-stimulating hormone (TSH)-like activity, often causing a transient decrease in maternal TSH levels during the first trimester. These adaptations help maintain adequate free T3 and T4 levels, which are essential for fetal neurodevelopment and metabolic regulation. Iodine deficiency during pregnancy can impair fetal development, underscoring the importance of thyroid hormone regulation during this critical period.”

[4]. In the method section, it is necessary to mention other independent variables such as age, number of pregnancies, prior thyroid illness, body weight, other pregnancy complications, etc.
Age was noted for all women, the number of previous pregnancies was available for few women and was not included in the analysis, prior thyroid illness was an exclusion factor, pre -pregnancy body weight and weight change in pregnancy were available for few women and were not included in the analysis, other pregnancy complications were not noted. In the revised version of the manuscript, we added however the presence of high levels of anti-TPO antibodes (anti-TPO antibody values >34 U/L were considered as being positive).

[5]. What were the inclusion and exclusion criteria?
The inclusion criterion included singleton pregnancy in women aged 18 and older while the exclusion criterion was medical history of thyroid disease and/or use of thyroid hormone before, during or after pregnancy. This was added in the revised version of the manuscript.

[6]. In the figure 2, what is PP?
PP denotes the Postpartum period. This has been clarified in the revised version of the manuscript.

[7]. Considering low and high levels of iodine, the correlation between iodine concentration and TSH or FT4 could be done. Which variable has the best correlation with iodine? TSH, FT4, or TFQI.

We checked the data for the three trimesters of pregnancy and the postpartum period and the only significant correlation that we noted was between FT4 and urinary iodine in the first trimester of pregnancy (r: +0.095, p: 0.05); all the other correlations for urinary iodine were not significant. We have added the following in the Results:

“A significant correlation was noted between FT4 and UI in the first trimester of preg-nancy (r: +0.095, p: 0.05); other correlations for UI with TSH, fT4 were not significant. …”

[8]. Consider discussing the limitations of the study.

We have added/changed the limitations of the study as follows:

“To acknowledge the limitations of our study we have to note the lack of data on Body Mass Index (BMI), weight change in pregnancy, TSH, fT4 and UI before pregnancy. Data on pregnancy complications were also not available. A more nuanced statistical analysis would be preferable for our data, however there are caveats: not all subjects had assessments for more than one trimester and accessible data on exact gestational age in weeks was currently available for a minority of women. Thus, we treated each measurement as being a separate one. The relationship between TFQI and UI can be confounded by several characteristics of the participants (such as maternal age and smoking); a more complex analysis with linear mixed models for example, could even adjust for such covariates.”

Reviewer 2 Report

Comments and Suggestions for Authors

Manuscript: Changes of central sensitivity to thyroid hormones vs urine iodine during pregnancy by Ilias et al.

               This study is crucial for several reasons, as thyroid hormones play an important role in the regulation of metabolism, growth and development during pregnancy. Maternal thyroid hormone levels change significantly to support foetal development, especially in the early stages when the foetus is dependent on maternal hormones. During pregnancy, the need for iodine increases as it is required for proper brain development and general foetal growth. Adequate iodine supply is crucial to prevent deficiency, which can lead to complications such as cognitive impairment and developmental disorders in the offspring.

               Although this is not a new topic, understanding these changes is critical to maternal and child health. Understanding the link between changes in thyroid hormone sensitivity and iodine levels is critical to preventing adverse pregnancy outcomes. This type of research can help identify at-risk populations, and while the research is not top news, it does reveal new links between hypothalamic-pituitary, thyroid hormones and iodine intake and is therefore necessary as an ongoing reminder and guide for public health policy to ensure adequate iodine intake during this important period of foetal development.

               However, I would like to ask you to elaborate on the equation used to calculate the TFQI in the Materials and Methods section, even if it is presented this way in other journals, this would be useful for the general reading public. This would make it easy to calculate this factor in a doctor's office and give an indication of how the levels in certain pregnant women relate to optimal levels of hypothalamic-pituitary and thyroid hormones and adequate iodine intake:

The TFQI was calculated to assess central sensitivity to thyroid hormones as follows: TFQI = cdfFT4 - (1 - cdfTSH), where cdf is the cumulative distribution function of the parameter [4].

Author Response

Responses to Reviewers’ comments

We thank the reviewers for their comments and suggestions. Please see below for our detailed report regarding the revised version of our manuscript

[Reviewer 2]
[1]. I would like to ask you to elaborate on the equation used to calculate the TFQI in the Materials and Methods section, even if it is presented this way in other journals, this would be useful for the general reading public. This would make it easy to calculate this factor in a doctor's office and give an indication of how the levels in certain pregnant women relate to optimal levels of hypothalamic-pituitary and thyroid hormones and adequate iodine intake: The TFQI was calculated to assess central sensitivity to thyroid hormones as follows: TFQI = cdfFT4 - (1 - cdfTSH), where cdf is the cumulative distribution function of the parameter.

We have added the following in the Conclusion section of the paper:

“The calculation of TFQI, as reported in the manuscript, could serve as an ancillary parameter for assessing iodine nutritional status during pregnancy and identifying pregnant women who may require iodine supplementation. The calculation of TFQI necessitates a – preferably – large size of data, so that the position, within the distribution of all the values in this dataset for a subject’s TSH and FT4 can be pinpointed. Thus, a tool, like a visual guide or a nomogram, using TFQI could be created to assess the probability of iodine adequacy in pregnancy. Additionally, assessing TFQI with io-dine intake in pregnant women could help determine the appropriate iodine dosage for each trimester, in conjunction with monitoring FT4 and TSH levels.”

Reviewer 3 Report

Comments and Suggestions for Authors

Dear Authors,

I have read the manuscript by Ilias et al. with great interest. The need for adequate iodine intake during pregnancy has been reported in various studies, however a deficiency in iodine intake for pregnant women is often present with consequences on fetal and child health and maternal status.

Detection of TFQI as reported in the manuscript could be a useful parameter to evaluate the nutritional iodine status during pregnancy and to find pregnant needing iodine supplementation. It could be interesting to correlate TFQI with pregnant iodine intake in order to consider the correct iodine dosage for the three trimesters beyond FT4 and TSH values. 

I think that a table with patients' characteristics could be useful, I have read that any information on their lifestyle (like smoking habits) are lacking but could be useful to report the data available about patients studied. 

Comments on the Quality of English Language

Minor revision for English text editing.

Author Response

Responses to Reviewers’ comments

We thank the reviewers for their comments and suggestions. Please see below for our detailed report regarding the revised version of our manuscript

[Reviewer 3].
Comments and Suggestions for Authors

[1]. Detection of TFQI as reported in the manuscript could be a useful parameter to evaluate the nutritional iodine status during pregnancy and to find pregnant needing iodine supplementation. It could be interesting to correlate TFQI with pregnant iodine intake in order to consider the correct iodine dosage for the three trimesters beyond FT4 and TSH values.

We agree with the suggestion and we have added the following in the Conclusion section of the paper:

“The calculation of TFQI, as reported in the manuscript, could serve as an ancillary parameter for assessing iodine nutritional status during pregnancy and identifying pregnant women who may require iodine supplementation. The calculation of TFQI necessitates a – preferably – large size of data, so that the position, within the distribution of all the values in this dataset for a subject’s TSH and FT4 can be pinpointed. Thus, a tool, like a visual guide or a nomogram, using TFQI could be created to assess the probability of iodine adequacy in pregnancy. Additionally, assessing TFQI with io-dine intake in pregnant women could help determine the appropriate iodine dosage for each trimester, in conjunction with monitoring FT4 and TSH levels.”

[2]. I think that a table with patients' characteristics could be useful, I have read that any information on their lifestyle (like smoking habits) are lacking but could be useful to report the data available about patients studied.

As explained in the article’s text some information is lacking from our dataset. However, anti-thyroid peroxidase positivity (over 34 U/mL) was available and is added to the Table, along with the results of the negative analysis of UI vs anti-TPO positivity.

[3]. Comments on the Quality of English Language
Minor revision for English text editing.

The article’s text has been checked and revised

Round 2

Reviewer 1 Report

Comments and Suggestions for Authors My comments were taken into consideration, thanks